

# Alleviation of cold stress impacts on grapes by the chitosan-salicylic acid nanocomposite (CS-SA NCs) application

Mohammad Ali Aazami[1], Lamia Vojodi Mehrabani[2], Mohammad Bagher Hassanpouraghdam[1], Farzad Rasouli[1], Gholam Reza Mahdavinia[3], Sona Skrovankova[4], Sezai Ercisli[5] and Jiri Mlcek[4]

[1] Department of Horticultural Science, University of Maragheh, Maragheh, Iran
[2] Department of Agronomy and Plant Breeding, Azarbaijan Shahid Madani University, Tabriz, Iran
[3] Polymer Research Laboratory, Department of Chemistry, University of Maragheh, Maragheh, Iran
[4] Department of Food Analysis and Chemistry, Tomas Bata University, Zlín, Czech Republic
[5] Department of Horticulture, Faculty of Agriculture, Ataturk University, Erzurum, Turkey

Corresponding authors
Mohammad Ali Aazami,
aazami58@gmail.com
Jiri Mlcek, mlcek@utb.cz

## ABSTRACT

Low temperature is a major abiotic stressor that limits the distribution of various fruit tree species worldwide. In this study, the effects of chitosan-salicylic acid nanocomposite (CS-SA NCs) treatment were evaluated on antioxidant enzyme activity, the antioxidant pool, and membrane stability indices in the grapevine cultivar 'Flame seedless' under cold stress conditions. Prolonged exposure to cold, compared to the control, led to a reduction in maximum fluorescence (Fm), variable fluorescence (Fv), the ratio of photochemical to non-photochemical use of light energy in photosystem II (Fv/F0), and the ratio of variable fluorescence to maximum fluorescence (Fv/Fm). Levels of chlorophylls, carotenoids, total soluble protein content, antioxidant enzyme activity, ascorbate, and glutathione activity significantly decreased with increasing cold stress duration. Electrolyte leakage, malondialdehyde, and hydrogen peroxide content increased by 75%, 60%, and 80%, respectively, after 16 hours of cold stress compared to the control. Nanocomposite treatment significantly enhanced antioxidant activity and stabilized membranes under cold stress by reducing electrolyte leakage and malondialdehyde release. Overall, CS-SA NCs act as a biological stimulant and can be effectively used to improve the physiological and biochemical responses of grapevines under cold stress. Further research is needed to gain a comparative understanding of various physiological responses, which will help guide the application of this nanocomposite in extension services and field production systems.

## INTRODUCTION

Low temperature is one of the main factors limiting fruit production and quality. While plants respond to variuos abiotic stresses in similar ways, each stressor can also trigger specific physiological and biochemical responses (*Zhang et al., 2022*). In addition to its direct effects, cold stress exerts indirect influences on plants. Elevated levels of reactive

oxygen species (ROS) molecules, including superoxide and hydrogen peroxide, significantly affect cellular activity and function (*Aazami, Mahna & Hasani, 2014*; *Hasanuzzaman, Hossain & Fujita, 2012*). The primary physiological impacts of cold stress include reduced respiration, changes in enzyme activity and hormone levels, disruption of photosynthetic electron transport chains, and increased production of reactive oxygen radicals (*Leuendorf, Frank & Schmülling, 2020*; *Ritonga & Chen, 2020*). Photosynthesis is the most vital process in plants (*Nishiyama & Murata, 2014*). Cold stress can impair its efficiency by reducing the turnover of the D1 protein in photosystem II (PSII) or by decreasing levels of photosynthetic pigments such as chlorophylls (*Lee et al., 2020*). Plant adaptation to low temperatures involves changes in the expression of numerous genes, proteins, and metabolites, as well as significant modifications in the content, composition, and organization of membrane lipids (*Miura & Furumoto, 2013*). The plasma membrane is the first cellular structure affected by cold stress, which alters its viscosity and represents the initial plant response at the cellular level (*Sharma, Sharma & Deswal, 2005*; *Wathugala, 2013*). Loss of membrane integrity due to low temperatures leads to a decrease in the ratio of saturated fatty acids (*Aazami et al., 2021a*; *Mahajan & Tuteja, 2005*). Exposure to cold stress in apples causes a marked increase in sugar levels and activates carbohydrate metabolism pathways. The effectiveness of plant defense responses to stress largely depends on the coordinated action of specific enzymes and the antioxidant action (*Xu et al., 2023*). Any imbalance in these components can result in severe dysfunction (*Hasanuzzaman et al., 2019*). Antioxidants play an essential role in maintaining the mentioned balance since these metabolites have the potential to react directly with different types of reactive oxygen species (ROS) molecules to efficiently scavenge excess ROS (*Hasanuzzaman et al., 2019*; *Spanò et al., 2017*).

Several studies have highlighted the role of salicylic acid (SA) in signal transduction pathways related to tolerance against biotic and abiotic stresses (*Miura & Tada, 2014*). Exogenous application of salicylic acid induces the expression of several stress-related proteins (*Aazami & Mahna, 2017*; *Miura & Tada, 2014*) and is essential for the development of systemic acquired resistance (SAR) (*Zhang et al., 2011*). SA regulates physiological processes such as photosynthesis, nitrogen and proline metabolism, glycine betaine synthesis, and enhances the antioxidant defense system, thereby improving stress tolerance (*Iqbal, Umar & Khan, 2015*). Its application prior to cold exposure influences lipid peroxidation, $H_2O_2$ levels, and antioxidant enzyme activity, depending on the timing of application (*Aazami et al., 2021a*). SA has been shown to improve cold resistance in crops such as corn, tomato, banana, red grape, cabbage, radish, cucumber, and barley (*Mutlu et al., 2016*). In grapes, SA mitigates frost stress by enhancing antioxidant enzyme activity, reducing cellular damage, and improving stress tolerance, with ascorbic acid being particularly effective (*Jalili et al., 2023*).

Chitosan also alleviates the effects of abiotic stress in plants (*Hidangmayum et al., 2019*). It promotes plant growth and enhances the uptake of water and essential nutrients, thereby improving ROS inhibition (*Sampedro-Guerrero et al., 2022*). Pretreatment with chitosan under salinity stress increases antioxidant enzyme activity and reduces malondialdehyde (MDA) levels, mitigating the negative effects of salinity in rice (*González et al., 2015*) and corn (*Al-Tawaha et al., 2018*). Application of chitosan nanoparticles (CS-NPs) enhances

cold stress tolerance in banana plants by reducing oxidative stress and promoting the accumulation of osmoprotectants such as soluble carbohydrates, proline and amino acids (*Wang et al., 2021a*). Recent studies have recommend CS-NPs to improve crop tolerance to various stress conditions, including salinity, drought, heavy metal toxicity, and heat (*Al-Tawaha et al., 2018*; *González et al., 2015*). In *Arabidopsis thaliana* treatment with chitosan-salicylic acid (CS-SA) resulted in greater root and rosette growth compared to treatment with free SA (*Sampedro-Guerrero et al., 2022*).

Grapevine (*Vitis vinifera* L.) is one of the most valuable fruit crops cultivated globally. Iran is the 11th largest grape producer globally, with an annual output of 1.94 million tons (*Epule et al., 2021*). In temperate regions, cold stress in vineyards reduces yield by damaging young shoots. While *V. vinifera* cultivars can tolerate temperatures as low as −25 °C during the winter season, they are vulnerable during the growing season, with temperatures between 0 °C and −4 °C capable of damaging or destroying green tissues and even entire plants (*Fennell, 2004*; *Mills, Ferguson & Keller, 2006*). Early spring frosts are a major limiting factor in grape production, sometimes causing up to 90% damage in vineyards (*Poling, 2008*).

To date, no studies have investigated the role of chitosan-salicylic acid nanocomposite (CS-SA NCs) in protecting grapevine plants against cold stress. Therefore, this study aims to evaluate the effects of foliar application of CS-SA NCs in mitigating the adverse impacts of cold stress in grapevine plants. We hypothesize that the application of CS-SA NCs enhance cold stress tolerance by reactivating defense mechanisms in grapevine plants.

## MATERIALS AND METHODS

### Plant material and growing conditions

In accordance with relevant institutional and national guidelines and legislation, uniform-sized cuttings of *Vitis vinifera* 'Flame seedless' cultivar were obtained from a local nursery. The cuttings were rooted in perlite and cultivated in a greenhouse maintained at 22–25 °C during the day and 18–20 °C at night. This experiment was conducted on biennial plants (2 years old). During acclimation, the plants were nourished using the hydroponic nutrient solution described by *Cramer & Läuchli (1986)*. A two-branch pruning method was applied to the young plants.

Foliar treatment with chitosan-salicylic acid nanocomposite (CS-SA NCs) (2 liters) was applied when the plants had developed 10–12 fully expanded leaves. Treatments were applied on plants 1, 6, and 12 h prior to cold stress exposure, while control plants were sprayed with distilled water (0.5 liter). Following each foliar treatment, the plants were transferred to cold storage for 1, 4, 8, and 16 h at +1 ± 0.5 °C; control plants were maintained at 22 °C. The experimental design included two factors, the first factor was CS-SA NCs application at four intervals (control, 1, 6, and 12 h before cold stress); the second factor was cold stress duration (1, 4, 8, and 16 h at +1 ± 0.5 °C, and control at 22 °C), in four replications. After cold treatment, leaf samples (from fully expanded leaves) were collected. Some were immediately transferred to the laboratory for analysis, while others were frozen in liquid nitrogen and stored at −80 °C until further evaluation.

## Synthesis of chitosan-salicylic acid nanocomposites (CS-SA NCs)

Low molecular weight chitosan (Mw = 100 kDa, DD = 85%, Purity = 97 wt%) was obtained from Dr. Mahdavinia company, Maragheh, Iran. Tripolyphosphate (TPP; Merck company, Germany) and used as a crosslinking agent to form chitosan nanoparticles. The synthesis method followed the gelation procedure described by *Ahmadi et al. (2018)*. To load salicylic acid (SA), 0.0138 g (0.1 mmol) of SA powder was dissolved in 1,000 mL of distilled water. Then, 1 mL (0.1 wt%) of acetic acid was added to the SA solution tofacilitate chitosan dissolution. To reach (SA)-loaded chitosan nanocomposites, 0.1 wt% of chitosan solution was prepared by adding 1 g of chitosan powder to the SA solution. After complete dissolution, 0.4 g of TPP was dissolved in 20 mL of distilled water and slowly added to the chitosan solution under continuous stirring (700 rpm). The resulting homogeneous SA-loaded chitosan nanocomposites were used for foliar applications without further modification.

## Chitosan-salicylic acid nanocomposite (CS-SA NCs) characterization

Transmission Electron Microscopy (TEM) analysis of the fabricated CS-SA NCs clearly illustrates the successful formation of well-defined octahedral structures, with particle sizes ranging from approximately 70 to 100 nm, indicating a uniform and controlled synthesis process (Fig. 1A). These findings were further supported by Dynamic Light Scattering (DLS) analysis, which validated the consistency of particle size distribution (Fig. 1B).

## Measurement of chlorophyll fluorescence

Chlorophyll fluorescence was measured in the most recently developed leaves under light-mode conditions using a portable fluorometer (PAM 2500-WALZ, Germany). These measurements were used to assess photosynthetic efficiency and the physiological response of plants to environmental conditions or treatments. The fluorometer provided precise readings of key parameters, including the maximum quantum yield of Photosystem II (PSII) and other fluorescence-related indicators.

## Electrolyte leakage

Electrolyte leakage was assessed by measuring the electrical conductivity (EC) of leaf samples, following the method described by *Nayyar (2003)*. Ten uniform leaves were placed in 20 ml of distilled water andshaken for 24 h at room temperature. The initial electrical conductivity 1 (EC1) was determined with an electrical conductivity (EC) meter. Then, the samples were autoclaved for 2 h at 120 °C, and the final electrical conductivity 2 (EC2) was recorded. Electrolyte leakage was calculated using the formula (*Nayyar, 2003*):

$$EC\,(\%) = (EC1/EC2) \times 100.$$

## Photosynthetic pigments

Chlorophylls (Chl a and Chl b), and carotenoids (CARs) were determined spectrophoto-metrically. Fresh leaf tissue (0.5 g) was ground using liquid nitrogen to preserve pigments and prevent degradation during the extraction process. The ground material was then suspended in 10 ml of 80% acetone as the extraction solvent, ensuring efficient solubilization of chlorophylls and carotenoids. The resulting mixture was thoroughly homogenized and

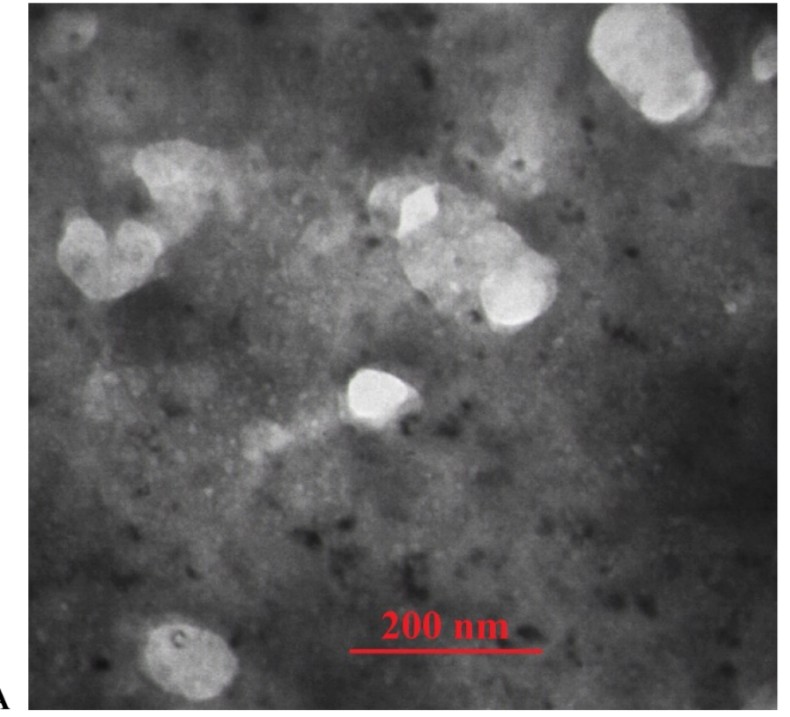

**A**

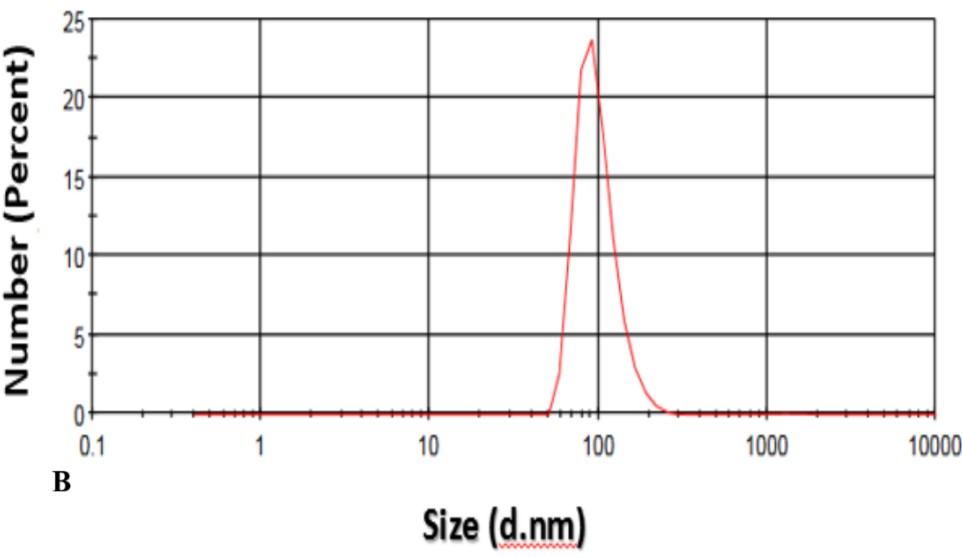

**B**

**Figure 1**  TEM image of sonochemical synthesis of CS-SA nanocomposite (A), and DLS analysis of CS-SA nanocomposites (B).

centrifuged to separate the supernatant containing pigments. The concentrations of Chl a, Chl b, and CARs were accurately quantified by measuring the absorbance of the pigment

extract at 663 nm, 645 nm, and 470 nm and expressed as mg g$^{-1}$ fresh weight (FW) by the below equations (*Arnon, 1949*).

    (1) *Chlorophyll a* $= [(12.7(A663)) - (2.69(A645))] \times v/1000w$
    (2) *Chlorophyll b* $= [(22.9(A645)) - (2.69(A663))] \times v/1000w$
    (3) *Carotenoids* $= [100(A470) + 3.27(Chl\ a) - 104(Chl\ b)]/227$.

## Malondialdehyde content

Grape leaf samples (0.2 g) were thoroughly homogenized in 5 mL of 1% trichloroacetic acid (TCA) and centrifuged at 12,000 g for 15 min to separate the solid and liquid phases. 1 mL of the resulting supernatant was mixed with 4 mL of a solution containing 20% TCA and 0.5% thiobarbituric acid (TBA), then heated for 30 min. Samples were immediately cooled on ice. Absorbance of the samples was subsequently measured using a T80+ spectrophotometer (China) at two specific wavelengths: 532 and 600 nm. MDA content was calculated and expressed as nmol g$^{-1}$ FW using the following formula (*Heath & Packer, 1968*):

$$MDA = [(A532nm - A600nm) \times 20]/155 \times 100.$$

## Hydrogen peroxide content

Frozen grape leaf tissue (0.2 g) was homogenized in two mL of 0.1% trichloroacetic acid (TCA) and centrifuged at 12,000 g for 15 min. The supernatant (0.5 mL) was mixed with 0.5 mL of phosphate buffer (10 mL, pH 7.0) and 1 mL of potassium iodide solution (1 M). The reaction produced a colored complex. The absorbance of the resulting solution was measured at 390 nm using a spectrophotometer. Hydrogen peroxide content was quantified using a standard $H_2O_2$ curve and expressed as mg g$^{-1}$ FW (*Velikova, Yordanov & Edreva, 2000*).

## Enzymatic extraction

To assess the enzymatic activities of catalase and guaiacol peroxidase, 0.5 g of grape leaf tissue was homogenized in liquid nitrogen. The homogenate was then mixed with 5 mL of cold phosphate buffer (pH = 7.5) containing 0.5 mL ethylenediaminetetraacetic acid (EDTA), which stabilizes enzymes and prevents metal ion interference. The mixture was centrifuged at 15,000 rpm for 15 min at 4 °C. The resulting supernatant, containing proteins and enzymes, was carefully collected and stored at −20 °C until further analysis (*Sairam, Rao & Srivastava, 2002*).

## Total soluble protein content

The reaction mixture for determining total soluble protein content consisted of three components: 100 µL of enzyme solution, 200 µL of Bradford reagent, and 700 µL of deionized water. The mixture was allowed to react for 2 min to ensure optimal complex formation between the reagent and amino acids, resulting in a color change proportional to protein concentration. Absorbance was measured at 535 nm using a spectrophotometer. Protein concentration was quantified using a standard bovine serum albumin (BSA) and expressed as mg g$^{-1}$ FW (*Bradford, 1976*).

## Catalase activity

To assess catalase activity, the enzyme extract was mixed with cold buffer and EDTA. The reaction was initiated by adding 0.5 mL of 75 mM hydrogen peroxide. Enzymatic activity was monitored by measuring the decrease in absorbance at 240 nm over 1 min, which corresponds to the breakdown of hydrogen peroxide into water and oxygen. The rate of absorbance change was used to calculate catalase activity, expressedas $\mu$mol min$^{-1}$ mg$^{-1}$ protein (*Dezar et al., 2005*).

## Ascorbate peroxidase (APX) activity

Grape leaf tissue (0.5 g) was homogenized in liquid nitrogen and suspended in 2 ml of 0.1 M phosphate buffer containing 0.5 mM EDTA, 5% polyvinyl pyrrolidone, and 2 mM ascorbate. After centrifugation, the supernatant was used for the enzymatic assay. The reaction mixture consisted of 250 $\mu$L of phosphate buffer (25 mM, pH 7.0), 0.1 mM ethylenediaminetetraacetic acid (EDTA), 10 $\mu$L of hydrogen peroxide (1 mM), 250 $\mu$L of reduced ascorbate (0.25 mM), 50 $\mu$L of the enzyme extract, and 190 mL of distilled water. The reaction was monitored spectrophotometrically at 290 nm over 1 min. APX activity was calculated using the extinction coefficient of 2.8 mM$^{-1}$ cm$^{-1}$, and expressed as $\mu$mol min$^{-1}$ mg$^{-1}$ protein) (*Stewart & Bewley, 1980*).

## Guaiacol peroxidase (GPX) activity

The GPX assay was performed using a reaction mixture containing 1 mL of phosphate buffer (100 mM, pH 7.0), 0.1 mM ethylenediaminetetraacetic acid (EDTA), 1 mL of guaiacol (15 mM), 1 mL of hydrogen peroxide (3 mM), and 50 $\mu$L of enzyme extract. The reaction was initiated by adding the enzyme solution, and the increase in absorbance at 470 nm was monitored spectrophotometrically for 1 min. GPX activity was calculatedusing an extinction coefficient of 26.6 mM$^{-1}$ cm$^{-1}$. The enzyme activity was expressed as $\mu$mol min$^{-1}$ mg$^{-1}$ protein (*Yoshimura et al., 2000*).

## Superoxide dismutase (SOD) activity

Grape leaf tissue (0.5 g) was homogenized in liquid nitrogen and mixed with 50 mM sodium phosphate buffer, 1 mM EDTA, and 2% polyvinylpyrrolidone (PVPP). After centrifugation, the supernatant was used for the enzymatic assay. SOD activity was assessed by its ability to inhibit the photoreduction of nitroblutetrazolium (NBT). The reaction mixture included 50 mL of 50 mM potassium phosphate buffer (pH 7.5) 75 $\mu$M nitroblutetrazolium, 13 $\mu$M methionine, 0.1 $\mu$M EDTA, and 4 $\mu$M riboflavin. This process was monitored spectrophotometrically at 560 nm, and SOD activity was expressed as $\mu$mol min$^{-1}$ mg$^{-1}$ protein (*Giannopolitis & Ries, 1977*).

## Glutathione activity

Grape leaf tissue (0.2 g) was homogenized in 2 mL of 5% sulfosalicylic acid to extract cellular components, including glutathione. The homogenate was centrifuged at 15,000 rpm for 10 min. From the supernatant, 300 $\mu$L was collected and neutralized with 18 $\mu$L of 7.5 M triethanolamine. For quantification, 50 $\mu$L of the neutralized supernatant was mixed with 700 $\mu$L of 0.3 mM nicotinamide adenine dinucleotide phosphate (NADPH), 100 $\mu$L of

5,5′-dithiobis (2-nitrobenzoic acid) (DTNB), and 150 μL of 125 mM phosphate buffer (pH 6.5) containing 6.3 mM EDTA. The reaction was initiated by adding, 0.1 unit of glutathione reductase, which reduces oxidized glutathione (GSSG) to its reduced form (GSH), while oxidizing NADPH to $NADP^+$. Absorbance was measured at 412 nm. A standard curve using known concentrations of both reduced (GSH) and oxidized (GSSG) glutathione was used to quantify glutathione levels,. expressed as $\mu mol \ min^{-1} \ mg^{-1}$ protein (*Griffith, 1980*).

### Ascorbate activity

A 0.2 g portion of the grape leaf sample was homogenized in 1 metaphosphoric acid. The homogenate was centrifuged at 22,000 rpm for 15 min at 25 °C. To the supernatant, 150 μM phosphate buffer (pH 7.4) and 200 μL of distilled water were added. Subsequently, 400 μL of 10% trichloroacetic acid (TCA), followed by 400 μL of 44% phosphoric acid, 400 μL of 4% bipyridyl in 70% ethanol to form a colored complex, and 200 μL of 3% ferric chloride ($FeCl_3$) to catalyze the reaction. After vortexing, the samples were incubated at 37 °C for 1 hour to allow full color development. Absorbance was measured spectrophotometrically at 525 nm, corresponding to the absorption maximum of the ascorbate-bipyridyl complex. To determine total ascorbate (including both reduced ascorbate and oxidized ascorbate, dehydroascorbate, DHA), 100 μL of 10 mM dithiothreitol (DTT) was added to the reaction mixture. A standard curve was constructed using known concentrations of reduced ascorbate to ensure accurate quantification. Final results were expressed as $\mu mol \ g^{-1}$ FW (*Law, Charles & Halliwell, 1983*).

### Experimental design and data analysis

The experiment was conducted as a factorial based on a randomized complete block design (RCBD) with three replications. Analysis of variance (ANOVA) was performed using MSTAT-C version 6.2.1 (https://www.canr.msu.edu/afre/projects/microcomputer_statistical_package_mstat._1983_1985). Significant differences among means were evaluated with Duncan's multiple range test at $P < 0.05$ and $P < 0.01$. Standard deviations ($n = 3$) were calculated for all traits. Pearson's correlation and cluster dendrogram heat maps were generated using R software (version 4.1.1; June 2024), (URL https://cran.r-project.org/bin/windows/base/old/4.1.1/).

## RESULTS

### Chlorophyll florescence

Variance analysis revealed that the interaction between cold stress duration and CS-SA nanocomposite treatment significantly affected F0, Fm, Fv, Fv/F0, Fv/Fm and Y(II) ($P \leq 0.01$). As cold stress duration increased, F0 and Y(II) values rose in the 'Flame Seedless' cultivar, while Fm, Fv, Fv/F0, and Fv/Fm declined compared to the control. Application of CS-SA NCs under cold stress conditions longer than 4 h led to an increase in the Fv/Fm index, indicating reduced damage to the photosynthetic apparatus. 16 hours of cold stress increased F0 by 1.66 times compared to the temperature control. Pretreatment with CS-SA NCs 12 h before cold exposure under the same conditions. Cold stress for 16 h caused

**Table 1 Effect Nano (CS-SA) on FO, Fm, Fv, Fv/Fm, Fv/Fm, Y(II), under cold stress of 'Flame seedless' grapevine.**

| Time cold stress (hrs) | CS-SA (hrs) | F0 | Fm | Fv | Fv/F0 | Fv/Fm | Y (II) |
|---|---|---|---|---|---|---|---|
| | Control | 0.95 ± 0.021 g | 5.63 ± 0.056b | 4.67 ± 0.0.43b | 4.89 ± 0.041a | 0.83 ± 0.084a | 0.537 ± 0.011m |
| Control | 1 | 1.02 ± 0.045i | 4.68 ± 0.042 g | 3.66 ± 0.036ef | 3.57 ± 0.026f | 0.78 ± 0.059e | 0.846 ± 0.018l |
| | 6 | 1.23 ± 0.056ef | 6.27 ± 0.055a | 5.04 ± 0.027a | 4.07 ± 0.058d | 0.80 ± 0.067bcd | 0.551 ± 0.012kl |
| | 12 | 1.07 ± 0.064 h | 5.71 ± 0.026b | 4.64 ± 0.012b | 4.31 ± 0.029b | 0.81 ± 0.066b | 0.548 ± 0.015l |
| | Control | 0.98 ± 0.028ij | 5.18 ± 0.051de | 4.20 ± 0.041d | 4.27 ± 0.033bc | 0.81 ± 0.079b | 0.570 ± 0.009i |
| 1 | 1 | 1.01 ± 0.055i | 5.21 ± 0.058de | 4.19 ± 0.029d | 4.12 ± 0.051cd | 0.80 ± 0.054bc | 0.563 ± 0.011j |
| | 6 | 1.17 ± 0.061 g | 5.60 ± 0.049b | 4.43 ± 0.027c | 3.77 ± 0.048e | 0.79 ± 0.064de | 0.560 ± 0.006j |
| | 12 | 1.17 ± 0.082 g | 5.40 ± 0.038c | 4.22 ± 0.016d | 3.60 ± 0.029ef | 0.78 ± 0.052e | 0.557 ± 0.018jk |
| | Control | 1.30 ± 0.035d | 3.63 ± 0.038k | 2.32 ± 0.055i | 1.77 ± 0.018k | 0.63 ± 0.046j | 0.627 ± 0.019fgh |
| 4 | 1 | 1.24 ± 0.047e | 3.64 ± 0.061k | 2.39 ± 0.018i | 1.92 ± 0.055jk | 0.65 ± 0.074i | 0.630 ± 0.014fg |
| | 6 | 1.07 ± 0.081 h | 4.18 ± 0.041hi | 3.10 ± 0.046 g | 2.89 ± 0.062 g | 0.74 ± 0.062f | 0.624 ± 0.008gh |
| | 12 | 1.17 ± 0.059 g | 4.06 ± 0.043ij | 2.88 ± 0.031 h | 2.46 ± 0.034hi | 0.71 ± 0.038gh | 0.619 ± 0.016 h |
| | Control | 1.4 ± 0.046c | 4.26 ± 0.056 h | 2.85 ± 0.018 h | 2.03 ± 0.047j | 0.66 ± 0.052i | 0.653 ± 0.017de |
| 8 | 1 | 1.38 ± 0.081c | 4.97 ± 0.044f | 3.59 ± 0.045f | 2.60 ± 0.034 h | 0.72 ± 0.028 g | 0.660 ± 0.007cd |
| | 6 | 1.18 ± 0.074 g | 5.65 ± 0.048b | 4.47 ± 0.062c | 3.79 ± 0.029e | 0.79 ± 0.046cde | 0.645 ± 0.005e |
| | 12 | 1.19 ± 0.068fg | 5.4 ± 0.039c | 4.20 ± 0.022d | 3.52 ± 0.054f | 0.77 ± 0.034e | 0.636 ± 0.012f |
| | Control | 1.58 ± 0.038a | 4.02 ± 0.055j | 2.44 ± 0.041i | 1.54 ± 0.026l | 0.60 ± 0.028k | 0.681 ± 0.009a |
| 16 | 1 | 1.55 ± 0.065a | 4.69 ± 0.021 g | 2.13 ± 0.039 g | 2.01 ± 0.027j | 0.66 ± 0.018i | 0.687 ± 0.018a |
| | 6 | 1.47 ± 0.088b | 5.25 ± 0.061d | 3.77 ± 0.018e | 2.56 ± 0.039hi | 0.71 ± 0.034gh | 0.671 ± 0.017b |
| | 12 | 1.49 ± 0.079b | 5.08 ± 0.044ef | 3.58 ± 0.026f | 2.39 ± 0.052i | 0.70 ± 0.012 h | 0.667 ± 0.012bc |
| Significance | df | | | | | | |
| Cold stress | 4 | ** | ** | ** | ** | ** | ** |
| CS-SA | 3 | * | ** | ** | ** | ** | ** |
| A*B | 12 | ** | ** | ** | ** | ** | ** |
| Error | | 0.001 | 0.009 | 0.01 | 0.017 | 0.0002 | 0.0005 |
| CV (%) | | 2.73 | 1.91 | 2.71 | 4.19 | 1.35 | 1.47 |

**Notes.**
Mean with the same letters are not significantly different by Duncan grouping at ($P < 0.05$).
** Significant at $p \leq 0.01$.
* Significant at $p \leq 0.05$.

reductions of 28.59, 47.75, 68.5 and 27.71% in Fm, Fv, Fv/F0 and Fv/Fm, respectively, compared to the control. Pretreatment with CS-SA NCs, especially 12-hours prior to cold stress, improved Fm, Fv, Fv/F0 and Fv/Fm under 16-hour cold stress (Table 1).

## Photosynthetic pigments content

The interaction between cold stress duration and chitosan-salicylic acid nanocomposite treatment had no significant effect on chlorophyll a, chlorophyll b, or total chlorophyll content. However, the main effect of temperature was significant ($P \leq 0.01$). Increasing exposure time at +1 °C significantly reduced chlorophyll a, chlorophyll b, and total chlorophyll content. Foliar application of CS-SA NCs up to 12 h before cold stress improved chlorophyll content. Total chlorophyll content decreased by 69.86% after 16 h of cold stress (+1 °C) compared to the control. Application of CS-SA NCs 12 h before cold

treatment increased total chlorophyll content by 14.56% compared to the control. The highest carotenoids content was observed with foliar application of CS-SA NCs applied 12 h before cold stress under 8-hour exposure. The lowest carotenoid content was recorded in the 16-hour cold stress treatment without foliar application (Figs. 2A–2G).

## Membrane stability and signaling responses

Electrolyte leakage increased with longer cold stress exposure, while foliar application of CS-SA NCs significantly reduced leakage ($P \leq 0.05$). The highest electrolyte leakage (58.15%) occurred in the 16-hour cold stress treatment without foliar application, while the lowest (14.82%) was observed in the temperature control (22 °C). Total soluble protein content increased with cold stress up to 4 h, then significantly decreased At the level of $p \leq 0.05$ at 16 h. The highest total soluble protein content was observed in plants treated with CS-SA NCs 12 h before cold stress and exposed for 4 h. The lowest total soluble protein content was observed in the 16-hour cold stress treatment without foliar application. Malondialdehyde content increased with longer cold stress exposure. At 16 h, treatments with CS-SA NCs applied 1, 6, and 12 h before cold stress showed 2.44-, 3.23-, 4.33-, and 3.97- fold increases in MDA compared to the control, respectively. Hydrogen peroxide content also increased with cold stress duration. The highest hydrogen peroxide level was observed in the 16-hour cold stress treatment without foliar application, while the lowest was observed in plants treated with CS-SA NCs 12 h before stress and not exposed to cold (Figs. 3A–3D).

## Antioxidant enzymes activity

Activities of catalase (CAT), gluthatione reductase (GR), ascorbate peroxidase (APX), and guaiacol peroxidase (GPX) increased with cold stress up to 4 h, then declined with longer exposure (16 h). The highest CAT, GR, and GPX enzyme activities were observed in plants treated with CS-SA NCs 6 h before cold stress and exposed for 4 h, showing 7.55-, 4.8-, and 6.39-fold increases compared to the control, respectively. The lowest activities of CAT, GR, and GPX enzymes were recorded in the control group. APX activity peaked in plants treated with CS-SA NCs 1 h before cold stress and exposed for 1 and 4 h. The lowest APX activity was recorded in the 16-hour cold stress treatment without foliar application. The highest SOD activity was was recorded in plants treated with CS-SA NCs 6 h before cold stress and exposed for 4 h, while the lowest was found in the 16-hour cold stress treatment without foliar application (Figs. 4A–4E).

## Non-enzymatic antioxidant activity

Ascorbate (AsA) levels and AsA/Dehydroascorbate (DHA) ratio decreased with increasing cold stress duration, while DHA activity significantly increased. The highest activity of ascorbate (AsA) was observed in plants treated with CS-SA NCs 12 h before cold exposure and not subjected to cold stress, showing a 62% increase compared to the control. The lowest ascorbate activity was recorded in the 16-hour cold stress treatment without foliar application, which was 58% lower than the control. Similarly, reduced glutathione (GSH) levels and GSH/GSSG (oxidized glutathione) ratio declined with cold stress, while GSSG activity increased significantly with prolonged exposure. The highest glutathione (GSH)

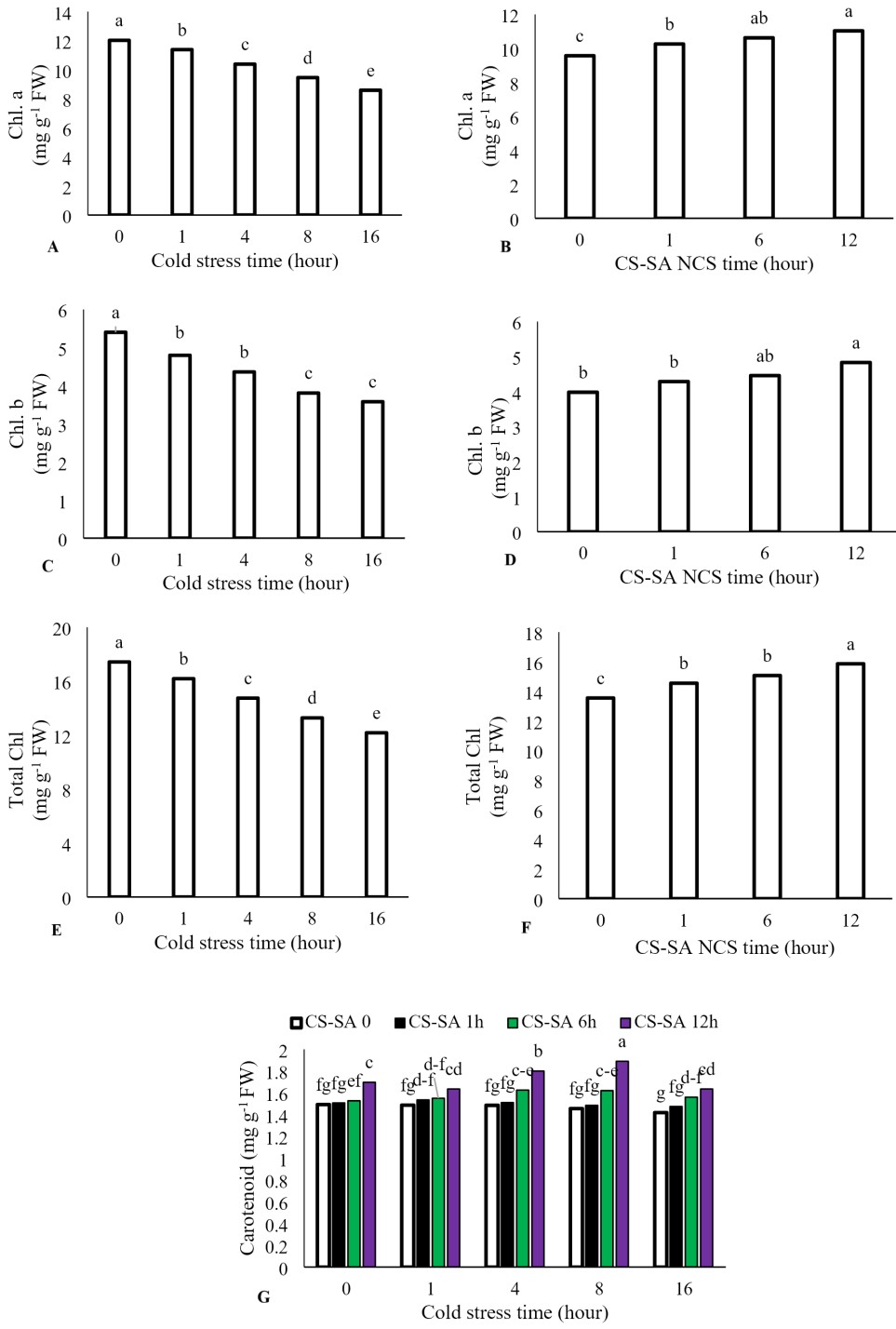

**Figure 2** The effects of cold stress time-course and CS-SA NCs on the chlorophyll a (A,B), chlorophyll b (C,D), total chlorophyll (E,F), and carotenoids content (G) of 'Flame seedless' grapevine. Similar letters on bars show no meaningful difference at 5% probability level by Duncan's Multiple Range Test. Data are mean ± SD ($n = 3$).

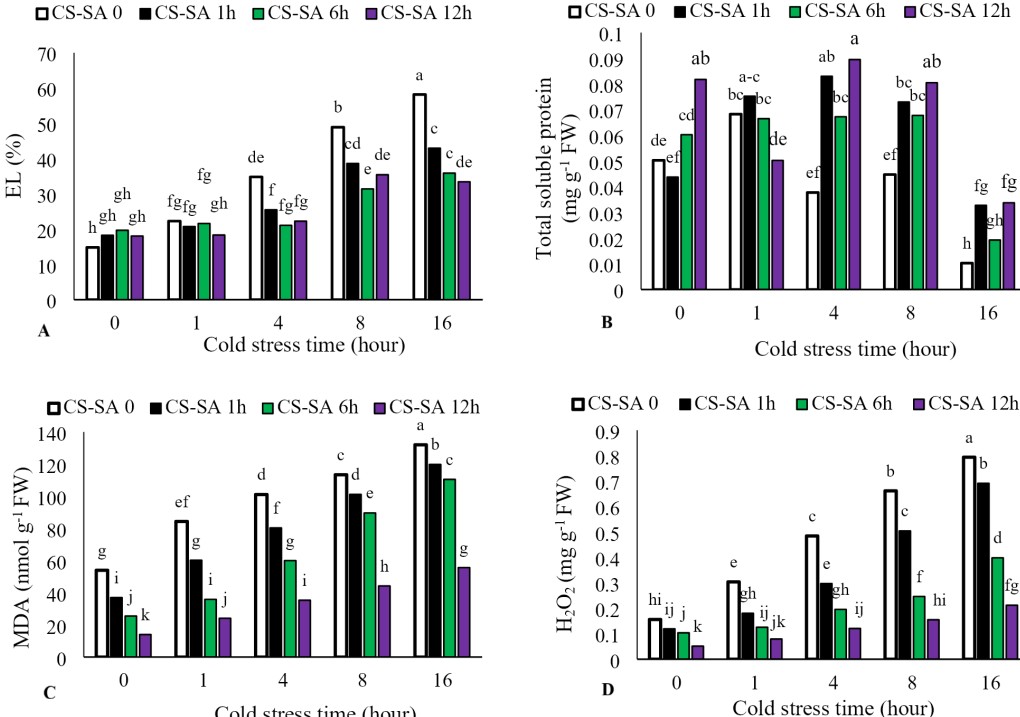

**Figure 3  The effects of cold stress time-course and CS-SA NCs foliar application on the electrolyte lakage (A), total soluble proteins (B), malondialdehyde (MDA) (C) and hydrogen peroxide (H2O2) (D) content of 'Flame seedless' grapevine.** Similar letters show no meaningful difference at 5% probability level by Duncan's Multiple Range Test. Data are mean ± SD ($n = 3$).

activity was observed in plants treated with CS-SA NCs 12 h before cold stress and not exposed to cold, showing a 61% increase compared to the control. The lowest glutathione activity was recorded in the 16-hour cold stress treatment without foliar application, showing a 55% decrease compared to the control (Figs. 5A–5F).

## Multivariate analysis of cold stress duration and CS-SA NCs foliar application in 'Flame seedless' grapevine

Pearson's correlation analysis of biochemical, enzymatic, and non-enzymatic antioxidant characteristics are shown in Fig. 5. Positive correlations were observed among Chl a, Chl b, total chl, carotenoids (CARs), enzymatic antioxidants, AsA, AsA/DHA, GSH, and GSH/GSSG. In contrast, photosynthetic pigments were negatively associated with MDA, GSSG, DHA, $H_2O_2$, and electrolyte leakage (EL). A significant positive correlation ($P \leq 0.05$) was observed among EL, GSSG, MDA, $H_2O_2$, and DHA. Electrolyte leakage was negatively correlated with enzymatic antioxidants, AsA, AsA/DHA, GSH, and GSH/GSSG (Fig. 6).

Heat map analysis of 'Flame seedless' grapevine plants responses to cold stress duration and CS-SA NCs treatment revealed that traits such as $H_2O_2$, MDA, and GSSG were positive associated with cold stress. Conversely, traits including photosynthetic pigments, protein content, and non-enzymatic antioxidants (AsA, GSH, and GSH/GSSG) showed negative

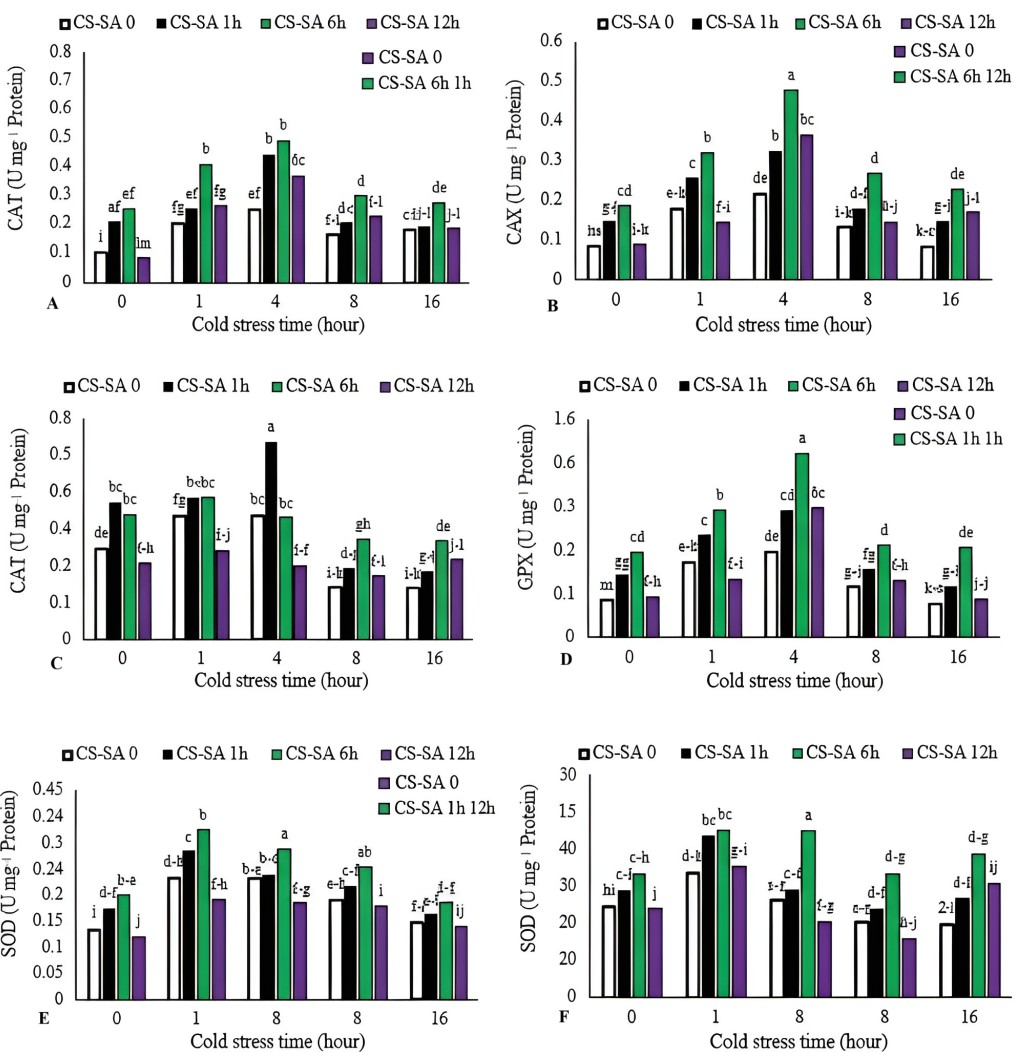

**Figure 4** The effects of cold stress time-course and CS-SA NCs foliar application on the CAT (A), GR (B), APX (C), GPX (D), and SOD (E) activity of 'Flame seedless' grapevine. Similar letters show no meaningful difference at 5% probability level by Duncan's Multiple Range Test. Data are mean ± SD (*n* = 3).

associations. The heat map indicated that CS-SA NCs application mitigated the effects of cold stress (Fig. 7A).

Cluster analysis and dendrograms identified three distinct groups under cold stress conditions. Group I contained: $H_2O_2$, GSSG, EL and MDA; group II included: CARs, DHA, CAT, AsA/DHA, AsA, GSH and GSH/GSSG, and group III consisted of Chl a, Chl b, total Chl, GPX, APX, GR, SOD and proteins content (Fig. 7A). Principal Component Analysis (PCA) confirmed the clustering observed in the heap map (Fig. 7B). Overall, cluster analysis revealed three treatment classes. Class I contained the plants treated with CS-SA NCs for 6 and 12 h and exposed to cold stress for 1 h, as well as untreated plants not exposed to cold stress. Class II contained plants exposed to cold stress for 4, 8, and

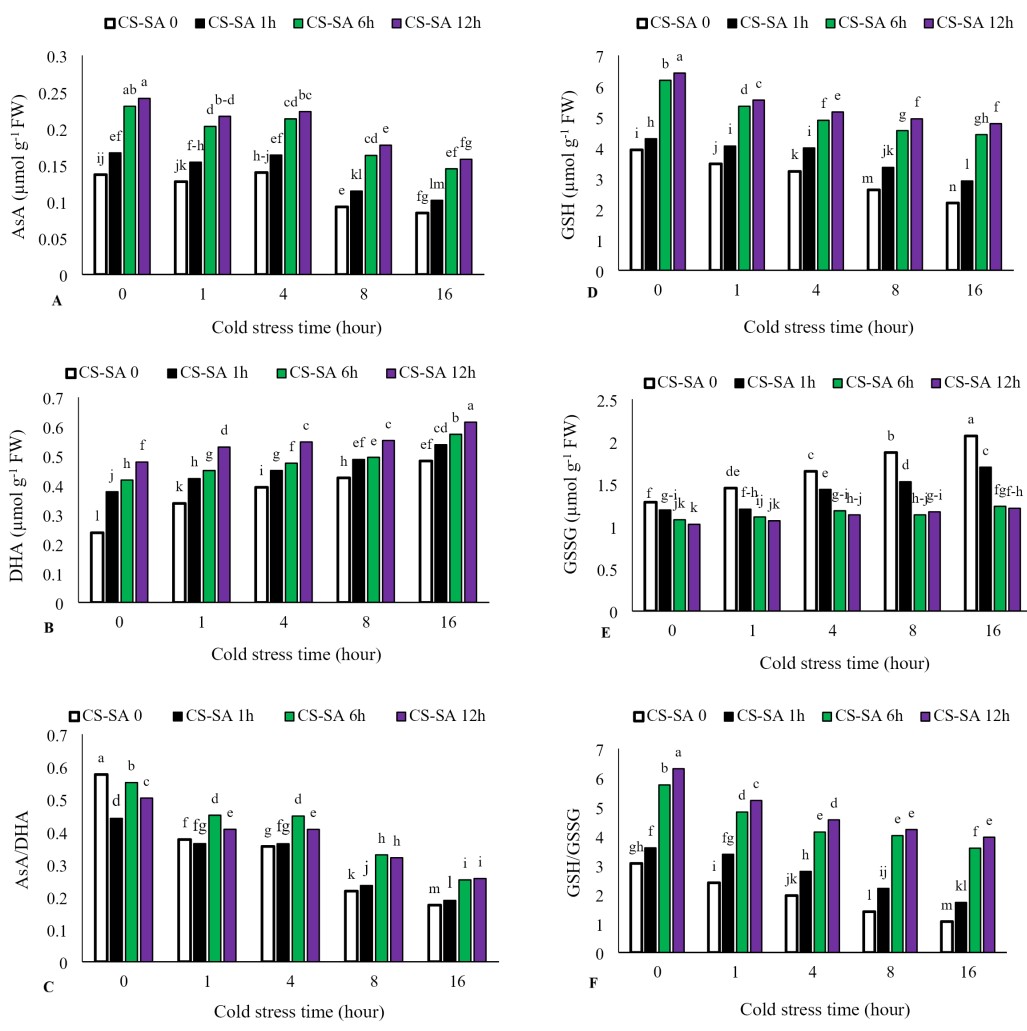

**Figure 5  The effects of cold stress time-course and CS-SA NCs foliar application on the AsA (A), DHA (B), AsA/DHA (C), GSH (D), GSSG (E), and GSH/GSSG (F) of 'Flame seedless' grapevine.** Similar letters show no meaningful difference at 5% probability level by Duncan's Multiple Range Test. Data are mean ± SD ($n = 3$).

16 h and treated with CS-SA NCs for 6 or 12 h. Finally, class III included plants exposed to cold stress treated with CS-SA NCs for 1 h, and untreated plsants under cold stress (Figs. 7A–7B).

## DISCUSSION

Chitosan nanocomposites act as elicitors, rapidly triggering plant defense responses upon application (*Stasińska-Jakubas & Hawrylak-Nowak, 2022*). Their interaction with plant surfaces and subsequent internalization can trigger signaling pathways involving salicylic acid (SA), jasmonic acid (JA), and abscisic acid (ABA), leading to the activation of transcription factors and downstream genes associated with abiotic stress tolerance (*Khan et al., 2022*). Transmission Electron Microscopy (TEM) analysis confirmed that

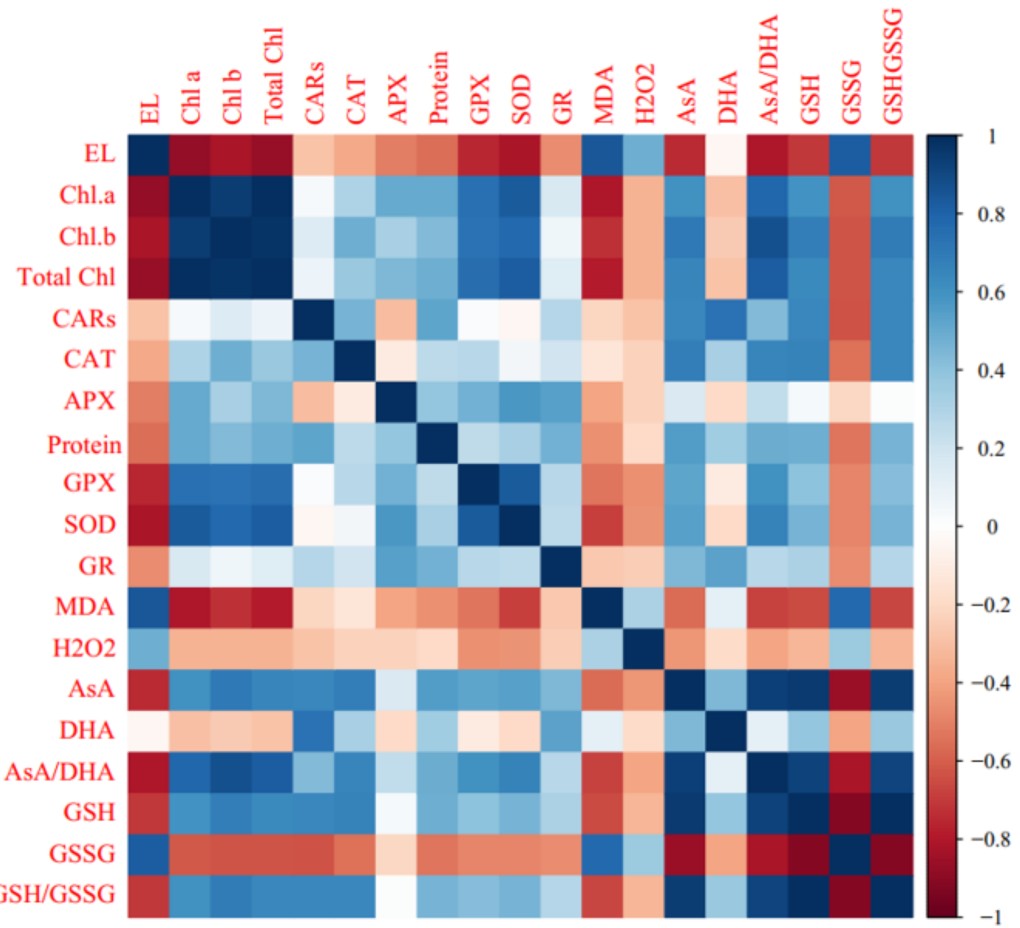

**Figure 6  Heat map of Pearson's correlation analysis for the 'Flame seedless' grape response to cold stress time-course and treated by CS-SA NCs.** Heat map representing Electrolyte leakage (EL), Chlorophyll a (Chl a), Chlorophyll b (Chl b), Total chlorophyll (Total Chl), Carotenoids (CARs), Malondialdehyde (MDA), H2O2 content, Total soluble proteins content, Guaiacol peroxidase (GPX) activity, Ascorbate peroxidase (APX) activity, Superoxide dismutase (SOD) activity, Gluthation reductase (GR), Ascorbate (AsA), Dehydroascorbate (DHA), AsA/DHA, Reduced glutathione (GSH), Oxidized glutathione (GSSG) and GSH/GSSG.

chitosan-salicylic acid (CS-SA) nanocomposite exhibit a uniform, spherical morphology with an average particle size of approximately 100 nm and good dispersion with minimal agglomeration. This nanostructure provides a high surface area and enhanced bioavailabilityenabling efficient interaction with plant tissues following foliar application (*Hong et al., 2021*). The nanoscale dimensions of CS-SA NCs facilitate rapid penetration through the leaf cuticle and uniform cellular distribution, allowing sustained release of both chitosan and SA. This enhances their protective effects, particularly during prolonged environmental stress (*Hoang et al., 2022*).

Low temperatures suppress photosynthesis through both stomatal and non-stomatal limitations (*Wang et al., 2021b*). PSII is the most sensitive and thermally responsive component of the photosynthetic apparatus. Chlorophyll fluorescence parameters are key
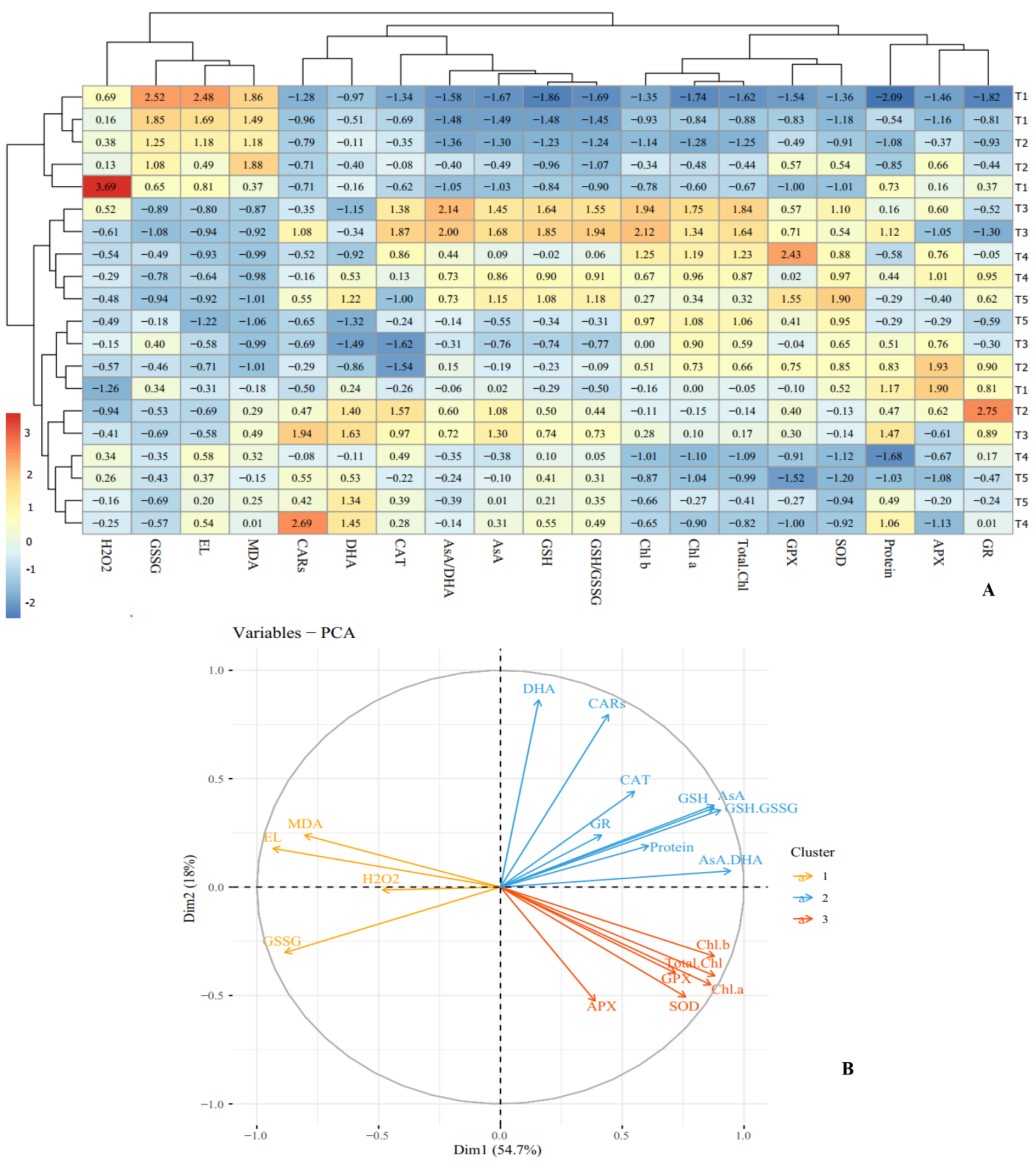

**Figure 7** Heat map (A) and loading biplot (B) of the enzymatic and non-enzymatic antioxidants pool and the biochemical traits changes in 'Flame seedless' grape under cold stress time-course treated by CS-SA NCs. Heat map representing Electrolyte leakage (EL), Chlorophyll a (Chl a), Chlorophyll b (Chl b), Total chlorophyll (Total Chl), Carotenoids (CARs), Malondialdehyde (MDA), H2O2 content, Total soluble proteins content, Guaiacol peroxidase (GPX) activity, Ascorbate peroxidase (APX) activity, Superoxide dismutase (SOD) activity, Gluthation reductase (GR), Ascorbate (AsA), Dehydroascorbate (DHA), AsA/DHA, Reduced glutathione (GSH), Oxidized glutathione (GSSG) and GSH/GSSG.

indicators of plant physiological status and stress responses (*Pradhan et al., 2019*). Cold stress increases F0 and decreases Fv/Fm, indicating disruption of light-absorbing pigments in photosystem II complexand reduced quantum efficiency. Photosystem II activity may be significantly impaired or halted under cold conditions, affecting chloroplasts, stromal carbon metabolism, and photochemical reactions in the thylakoid membrane (*Aazami*

*et al., 2021a*). Cold stress reduces Fv/F0 and Fv/Fm, with Fv/F0 being the most sensitive component of the photosynthetic electron transport chain. The Fv/Fm parameter is a reliable indicator of photoinhibition and damage to the photosynthetic apparatus, oftendetectable before visible symptoms appear (*Akhter et al., 2021*). Stress impairs enzyme function and reduces the activity of water-splitting complexes and electron transport chains, leading to decreased Fv/Fm (*ALKahtani et al., 2020*). Previous studies have shown that SA effects on photosynthetic efficiency are concentration-dependent under both optimal and stressful conditions (*Wang et al., 2021b*). Chitosan is known for its biopolymer film-forming properties, which enhance membrane stability by reinforcing lipid bilayers and reducing ion leakage, especially under low-temperature conditions (*Yu et al., 2022*). The nanoscale dimensions of CS-SA nanocomposite's improve cellular uptake and bioavailability, enabling effective delivery of stabilizing agents directly to chloroplasts (*Polyakov et al., 2023*). This results in preservation of thylakoid structure and a significant reduction in F0, as demonstrated by the 69.5% decrease in F0 with nano CS-SA pretreatment.

A significant decrease in chlorophyll a, chlorophyll b, and total chlorophyll levels was observed underprolonged low-temperature stress. This reduction is likely due to damage caused by free radicals. Alterations in nitrogen metabolism may also contribute to chlorophyll loss, as thesechanges affect the synthesis of compounds such as proline, which help regulate osmotic pressure (*Aazami et al., 2021a*). Chlorophyll biosynthesis is a temperature-sensitive process and is commonly used as a quantitative indicator of cold sensitivity across different plant species (*Hidangmayum et al., 2019*). The decline in chlorophyll content in plants under environmental stress may be attributed to changes in protein content and composition, alterations in the other pigments, or increased chlorophyllase activity (*Fozouni, Abbaspour & Baneh, 2012*). Reduced levels of chlorophyll b and a have been reported under cold stress also in watermelon (*Sayyari et al., 2013*). Application of salicylic acid under cold stress conditions has been shown to enhance chlorophyll concentration, likely due to an improved antioxidant defense mechanism (*Aazami, Mahna & Hasani, 2014*). Spraying plants with chitosan (250 mg L$^{-1}$) significantly increased chlorophyll and total carbohydrate content (*Farouk & Amany, 2012*; *Sayyari et al., 2013*). Chitosan treatment also facilitates access to amino acids and related proteins, which stimulate chlorophyll biosynthesis (*Zhang et al., 2011*). Additionally, salicylic acid-induced increases in carotenoid levels appear to enhance plant resistance to oxidative damage (*Fozouni, Abbaspour & Baneh, 2012*). CS-SA nanocomposites, due to their nanoscale properties, improve nutrient uptake and internal plant transport, ensuring the availability of essential elements such as magnesium and nitrogen for pigment biosynthesis (*Sen & Das, 2024*). The present study demonstrated that foliar application of chitosan-salicylic acid nanocomposites (CS-SA NCs) significantly reduced cold-induced electrolyte leakage in grapevine leaves, particularly when applied prior to stress exposure. Cold stress disrupts membrane function, leading to the leakage of electrolytes from cells (*Aazami, Mahna & Hasani, 2014*). Measuring electrolyte leakage is a reliable method for assessing membrane integrity and permeability following environmental stress, including cold (*Wang & Li, 2006*). In watermelon, salicylic acid application under cold stress reduced

leaf electrolyte leakage (*Sayyari et al., 2013*). Salicylic acid mitigates the harmful effects of stress and helps maintain membrane integrity by reducing electrolyte leakage (*Aazami, Mahna & Hasani, 2014*). Foliar application of salicylic acid to grape cuttings has been shown to enhance tolerance to cold and heat stress by reducing ion leakage and preventing cell membranes peroxidation (*Wang & Li, 2006*). Chitosan nanocomposites stabilize cell membranes by minimizing electrolyte leakage and preserving the structural and functional integrity of lipids under stress. This effect is attributed to chitosan's ability to bind with membrane phospholipids and modulate membrane fluidity, thereby preventing cellular dehydration and ion imbalance during stress episodes (*Balusamy et al., 2022*). The nanoscale size of the CS-SA particles (typically <100 nm, depending on synthesis method) provides a significantly higher surface area-to-volume ratio compared to bulk materials (*Nofal et al., 2024*). This ultra-small size facilitates the penetration through the leaf cuticle and entry into apoplastic and symplastic pathways (*Sembada & Lenggoro, 2024*). Furthermore, the uniform particle size ensures even distribution across the leaf surface, maximizing contact and enabling rapid uptake by epidermal cells (*Hu et al., 2020*). The positive charge of chitosan molecules promotes strong electrostatic interactions with negatively charged phospholipids in the plasma membrane. At the nanoscale, these interactions are further intensified, forming a protective biofilm-like layer around membrane surfaces (*Katiyar, Hemantaranjan & Singh, 2015*), which stabilizes the lipid bilayer, reduces membrane permeability, and effectively decreases ion efflux (electrolyte leakage) under stress (*Wagatsuma, 2017*).

Our results indicate that low-temperature treatment induced oxidative stress by increasing the production of hydrogen peroxide and malondialdehyde. Foliar application of SA and other stimulants enhanced cold tolerance and reduced the accumulation of $H_2O_2$, MDA, and superoxide radicals in roots and leaves under cold stress (*Aazami, Mahna & Hasani, 2014*; *Miura & Tada, 2014*). Both chitosan and salicylic acid, especially in nano-form, upregulate key antioxidant enzymes such as CAT, APX, and POD, which directly catalyze the breakdown of $H_2O_2$ into water and oxygen (*Sen & Das, 2024*). Nanoscale delivery enhances cellular signaling and transcriptional activation of these enzymes, resulting in faster and more efficient $H_2O_2$ detoxification (*Yadav et al., 2024*).

The protective effects of SA primarily stem from its ability to neutralize stress-induced free radicals, preventing damage to unsaturated fatty acids and reducing membrane permeability. This helps preserve cellular integrity and maintain normal metabolic functions during stress (*Borsani, Valpuesta & Botella, 2001*). Our findings align with those of *Wang et al. (2009)*, who reported reduced MDA levels following salicylic acid treatment. Several studies suggest that $H_2O_2$ acts as a signaling molecule that initiates protective responses in plants against stress. Another mechanism by which salicylic acid confers protection is trough the enhancement of hydrogen peroxide levels, a typical response of various biochemical pathways triggered by stress (*Miura & Tada, 2014*). Induced hydrogen peroxide activates calcium-dependent channels (*Wang et al., 2012*). Chitosan and its derivatives also reduce MDA levels and stabilize membranes, contributing to stress tolerance (*Wang et al., 2021a*). Environmental stresses significantly influence antioxidant enzyme activity and can affect protein biosynthesis, accumulation, and function (*Zonouri, Javadi*

& Ghaderi, 2014*). Salicylic acid plays a key role in modulating apoplastic proteins and antioxidant enzymes associated with cold stress tolerance (*Taşgín, Atící & Nalbantoğlu, 2003*). Reductions in protein synthesis or denaturation may be linked to changes in antioxidant activity (*Aazami et al., 2021a*).

The enzyme GPX is active in the cytosol and plays a key role in removing hydrogen peroxide (*Aazami, Rasouli & Panahi Tajaragh, 2021b*). Its activity varies depending on the cultivar; in cultivars tolerant to cold and other environmental stresses, GPX activity is typically higher than in sensitive ones. This increased activity suggests that GPX is a dominant enzyme in managing excess hydrogen peroxide in tolerant cultivars. CAT is another essential enzyme involved in scavenging hydrogen peroxide and mitigating its destructive effects on peroxisomes, glyoxysomes, and mitochondria. It also converts hydrogen peroxide generated by SOD into water. Reduced catalase activity under low-temperature stress has been reported in various plant species (*Aazami et al., 2022*; *Zhang et al., 2011*). The induction of CAT activity in response to biotic and abiotic stressors is well documented across many plant species (*Mutlu, Atici & Nalbantoglu, 2009*; *Taşgín, Atící & Nalbantoğlu, 2003*). CAT is particularly effective in removing $H_2O_2$ from the apoplast (*Kisa et al., 2021*). APX enzyme catalyzes theconversion of hydrogen peroxide into water in chloroplasts, cytosols, mitochondria, and peroxisomes of plant cells, and maintaining the hydrogen gradient required for various cellular reactions (*Kisa et al., 2021*; *Ritonga & Chen, 2020*). Increase activity of CAT, APX, and SOD enhances cold tolerance in plant species (*Yang, Wu & Cheng, 2011*). APX activity peaks following salicylic acid treatment, which elevates levels of AsA and $H_2O_2$ (*Aazami, Mahna & Hasani, 2014*; *Wang & Li, 2006*). In grapes, salicylic acid treatment improves tolerance to cold and heat stress by increasing AsA levels and subsequently boosting APX activity (*Wang & Li, 2006*).

APX neutralizes $H_2O_2$ by using the AsA cycle as an electron donor (*Hasanuzzaman et al., 2019*). GR converts glutathione oxide to its reduced form (GSH) using NADPH in the glutathione-ascorbate cycle. Enhanced GR enzyme activity helps maintain a favorable GSH/(GSH + GSSG) ratio, improving cellular defense mechanisms and significantly increasing plant tolerance to environmental stress (*Sohag et al., 2020*). SA-induced cold tolerance in maize and cucumber plants may be linked to increased glutathione reductase and peroxidase activities, indicating that salicylic acid influences antioxidant enzyme activity and hydrogen peroxide metabolism (*Sayyari et al., 2013*). Elevated SOD activity can lead to increased levels of superoxide anions and $O^{-2}$ radicals, which are generated viaNADPH oxidase in plasma membranes and subsequentlyconverted to $H_2O_2$ by SOD mediation (*Aazami et al., 2022*). Pretreatment with NaHS has been shown to enhance SOD activity in grape seedlings under low-temperature stress. (*Fu et al., 2013*). Studies on maize and grape have demonstrated that foliar application of salicylic acid under stress conditions increased superoxide dismutase and peroxidase activities (*Wang & Li, 2006*; *Wang et al., 2012*). *Kumaraswamy et al. (2019)* reported that chitosan and SOD can scavenge superoxide anions. Chitosan's neutralizing potential is attributed to its hydroxyl and amino groups, which effectively react with ROS (*Hidangmayum et al., 2019*). Exposure to salinity also elevates enzymatic antioxidant levels (*Aazami et al., 2022*). Chitosan-treated plants, playing a crucial role in mitigating environmental stress through enhanced antioxidant enzyme

activity and function (*Rabêlo et al., 2019*). Chitosan nanocomposites have been shown to upregulate the expression and activity of key antioxidant enzymes, including superoxide dismutase (SOD), catalase (CAT), ascorbate peroxidase (APX), and peroxidases (POD) (*Qu et al., 2019*). These enzymes work synergistically to neutralize ROS, conferring cellular protection and reducing oxidative damage to lipids, proteins, and nucleic acids (*Fujita & Hasanuzzaman, 2022*). Ascorbateis the most abundant antioxidant in plant cells and is found in all intracellular organs and apoplastic space, with average concentration ranging from 2 to 25 mM. In chloroplast stroma, its concentration may exceedcthis range. Ascorbate is oxidized by oxygen, superoxide, singlet oxygen, and $H_2O_2$ to form monodehydroascorbate reductase (MDHA), which eventually decomposes into ascorbate and dehydroascorbate (*Gill & Tuteja, 2010*). Ascorbate peroxidase (APX) assists ascorbate in purifying $H_2O_2$. Ascorbate can react directly with hydroxyl, superoxide, and singlet oxygen radicals. Due to its role in photosynthesis, AsA concentration is particularly high in chloroplasts, where it removes $H_2O_2$ generated during the oxygen reduction reaction in photosystem I (Mehler reaction) (*Hidangmayum et al., 2019*; *Tavanti et al., 2021*). Among the factors involved in the ascorbate oxidation and its conversion from reduced to oxidized form is activity of ascorbate peroxidase enzyme, which reduces the superoxide radicals and singlet oxygen levels. This enzyme controls the levels of alpha-tocopherol too (*Miret & Müller, 2017*). Reduced AsA content has been reported as a response to abiotic stresses (*Gill & Tuteja, 2010*). At the nanoscale, CS-SA NCs exhibit heightened surface reactivity, improving physical and chemical interactions with plant cellular structures. These interactions increase uptake and more effectively modulate signal transduction pathways compared to bulk materials (*Yadav et al., 2024*). Consequently, the enzymatic activities of APX, GPX, SOD, and GR are synergistically enhanced, providing superior protection against oxidative stress (*Rao et al., 2025*).

Glutathione regulates gene expression and mitigates herbicide toxicity by binding and chelating heavy metals *via* precursor glutathione S-transferase (*Hasanuzzaman et al., 2017*; *Tavanti et al., 2021*). Antioxidants such as glutathione also serve as a coenzyme for antioxidant enzymes such as glutathione reductase (*Israr & Sahi, 2006*). Increased expression and activity of antioxidants such as glutathione and ascorbate help maintain high ASA/(ASA + DHA) and GSH/(GSH + GSSG) ratios, preserving the structure and function of vital biomolecules (*Tavanti et al., 2021*). Glutathione is a key indicator of stress tolerance in plants and plays multiple roles in plant defense against diverse types of stressors (*Hasanuzzaman et al., 2017*; *Yu et al., 2020*).

Stress-induced reductions in GSH concentration have also been reported (*Gill & Tuteja, 2010*). Prolonged cold stress reduces glutathione reductase activity, and the balance between glutathione production and consumption is critical for its antioxidant properties (*Lefèvre et al., 2010*). GR and GSH regulate the components of the ascorbate-glutathione cycle. During $H_2O_2$ detoxification, GSH (reduced form) is converted to GSSG (oxide form), and then regenerated by GR-dependent NADPH activity. High GR activity helps maintain glutathione reserves (*Gill & Tuteja, 2010*; *Tavanti et al., 2021*). Nanoscale delivery systems provide higher local concentrations of elicitors at target sites, accelerating and amplifying the increase in GSH levels and the GSH/GSSG ratio, a key indicator of redox status (*Elena*

*et al., 2019*). Finally, the high surface reactivity and interaction potential of nanoparticles enable coordinated activation of antioxidant networks, enhancing the interplay between enzymatic (*e.g.*, SOD, CAT, APX) and non-enzymatic (GSH) defense mechanisms (*Singh et al., 2024*).

## CONCLUSION

Low-temperature stress is one of the major limiting factors for grapevine cultivation in many temperate regions. Cold conditions disrupt electron transfer chain, leading to elevated levels of reactive oxygen species (ROS). The plant's natural defense mechanism involves enhancing the activity of antioxidant enzymes and increasing antioxidants levels to mitigate ROS damage. The 'Flame Seedless' grapevine cultivar demonstrated strong potential for ROS suppression, particularly through elevated activities of ascorbate and glutathione. The application of salicylic acid-chitosan nanocomposite significantly improved tolerance to low-temperature stress. The nanocomposites induced acquired resistance by increasing $H_2O_2$ leves as a signaling molecule and strengthening the plant's defense system. Prolonged exposure to cold stress (8 and 16 h) resulted in excessive ROS production, triggering lipid peroxidation and cellular damage. However, the use of nanoparticles enhanced the enzymatic antioxdant defense system, helping to prevent maintain cellular stability and prevent damage. Further in-depth studies are necessary to compare the physiological responses of different grapevine cultivars. Such research will support the development of practical recommendations for the use of CS-SA nanocomposites in extension services and by pioneering growers in the field.

## ACKNOWLEDGEMENTS

The authors wish to thank the University of Maragheh, Iran for providing the samples and organizing the writing of this article.

### Funding
The authors received no funding for this work.

### Competing Interests
The authors declare there are no competing interests.

### Author Contributions
- Mohammad Ali Aazami conceived and designed the experiments, performed the experiments, prepared figures and/or tables, and approved the final draft.
- Lamia Vojodi Mehrabani performed the experiments, prepared figures and/or tables, and approved the final draft.
- Mohammad Bagher Hassanpouraghdam analyzed the data, prepared figures and/or tables, and approved the final draft.

- Farzad Rasouli conceived and designed the experiments, analyzed the data, prepared figures and/or tables, and approved the final draft.
- Gholam Reza Mahdavinia analyzed the data, prepared figures and/or tables, and approved the final draft.
- Sona Skrovankova conceived and designed the experiments, authored or reviewed drafts of the article, and approved the final draft.
- Sezai Ercisli performed the experiments, authored or reviewed drafts of the article, and approved the final draft.
- Jiri Mlcek performed the experiments, authored or reviewed drafts of the article, and approved the final draft.

## Data Availability

The raw data is available in the Supplemental File.

## Supplemental Information

Supplemental information for this article can be found online at http://dx.doi.org/10.7717/peerj.20368#supplemental-information.

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
