# Peer review of "Alleviation of cold stress impacts on grapes by the chitosan-salicylic acid nanocomposite (CS-SA NCs) application"

_PeerJ, doi:10.7717/peerj.20368_

## Round 0.1 · original submission · Major Revisions

· Academic Editor

Major Revisions

**Language Note:** PeerJ staff have identified that the English language needs to be improved. When you prepare your next revision, please either (i) have a colleague who is proficient in English and familiar with the subject matter review your manuscript, or (ii) contact a professional editing service to review your manuscript. PeerJ can provide language editing services - you can contact us at [email protected] for pricing (be sure to provide your manuscript number and title). – PeerJ Staff

Reviewer 1 ·

Basic reporting

The manuscript Alleviation of cold stress impacts on grapes by the chitosan-salicylic acid nanocomposite (CS-SA NCs) application is interesting and meets the standards to be published, however it can be improved by making adjustments to it.

Experimental design

Complement by adding the number of experimental units used to carry out the experiment

Validity of the findings

No comment

Additional comments

METHODOLOGY:
*Growing conditions: mention the age of the plants before being subjected to stress.
*Mention the volume used for foliar application of the nanocompound and the water used as a witness.
*In the methodology section, the characterization of the nanocomposite under TEM analysis is added, describing its size and distribution; the impact of these characteristics on the results obtained is added to the results and discussion section.
*Electrolyte leakage: Improve the description of the methodology by adding the weight or size of the plant material used for its implementation. In addition, the instrumentation used for EC measurement and the implementation of the methodology.
*The determination of photosynthetic pigments, Malondialdehyde content, and Hydrogen peroxide content was performed using a standard or an equation? (Mention this).
*It is mentioned that the enzyme extract is used only for the catalase and guaiacol peroxidase enzymes. Add the extract used to calculate the total soluble protein content, APX, and SOD.

DISCUSSION
*Improve the discussion by adding more specific information on how CS-SA influences the results obtained. The effect of chitosan is not ruled out, as it is mentioned very briefly.

Reviewer 2 ·

Basic reporting

Nowadays, extreme climatic conditions pose a great burden to farmers in agriculture and horticulture worldwide. Perennial crops planted in permanent locations are particularly at risk. Grapes are often exposed to cold stress in continental climates, which is difficult to protect against due to the perennial nature of the plant. The authors offer an alternative to this challenge, so the choice of topic is timely and of great interest.

The use of English makes the manuscript clear and unambiguous.

The introduction is well-structured, clearly outlines the problem and details the alternative solution. The references are relevant, in content and quantity they correspond to the level expected from a publication of this quality, but they are often old. More recent references are also available in the areas of abiotic stress, salicylic acid, and chitosan.

Experimental design

The planned experiment basically meets the requirements of the field, but contains some methodological problems (see detailed review suggestions).

The evaluation of the results often contains illogicalities and contradictions (see detailed review suggestions).

The figures and tables are clear, well-structured, interpretable, and suitable for presenting the desired results.

Validity of the findings

The discussion subsection clearly and well-formulatedly supports the results, and the authors compare their own results with relevant literary references. It would be good if the authors also cited more new literature in this section.

Additional comments

My suggestions for improving the quality of the manuscript are the following:

Line 24: The abbreviations in the abstract (Fm, Fv, Fv/F0 and Fv/Fm) must be written out as full words when they first appear.

Line 48: The term ROS must be written out as full words. ROS is not listed in the abbreviation list either.

Lines 79-84: Where does this statement come from? There is no literary reference after it. It also contains the abbreviated names of the antioxidant enzymes, but their full names are not written out.

Line 99: CH-NP is also described only in abbreviated form.

Line 101: The full term is missing next to the abbreviation CS:SA.

Line 122: "we tried to feed" this is a scientific manuscript, not a work of fiction. Correct: we fed.

127-129. lines: "After each foliar treatment, the plants were transferred to the cold storage for 1, 4, 8, and 16 hours at a temperature of +1± 0.5°C and the controls at a temperature of 22°C."
This can be considered a serious methodological error. Why were the untreated control plants not also exposed to the cold effect? Without this, the effects of the treatments cannot be compared.

Lines 297-298: the authors state that they found a significant difference, but they do not state at what significance level? This is questionable because their statistical tests were performed at two levels of significance.

Lines 302-305: "Cold stress of 16 hours enhanced F0 by 1.66
times compared to the control. Application of nano CS-SA, 12 hours before cold stress for 16 hours compared to the no-use of nano CS-SA in the same conditions of cold stress (16 hours) caused a 69.5% reduction in F0."
The two sentences completely contradict each other.

310-310. lines: "The interaction effect of cold stress time-course and chitosan-salicylic acid nanocomposite treatments on chlorophyll a, chlorophyll b, and total chlorophyll content was insignificant. With
the increasing exposure time at +1°C, the chlorophyll a, chlorophyll b, and total chlorophyll content significantly decreased."
The two sentences completely contradict each other.

318-319. lines: "The least carotenoids content was observed in 16-hrs
cold treatment without foliar application."
If such a treatment existed, why did the material and method section refer to plants that were not exposed to cold stress as controls?

lines 321-323: "Electrolyte leakage declined with increasing cold stress exposure time. The highest percentage of electrolyte leakage (58.15%) was observed in 16-hour cold stress without foliar application, and the least recorded data for electrolyte leakage (14.82%) was observed in control."

The two sentences are completely contradictory.

lines 343-344: "The lowest APX activity was recorded
in 16-hour cold treatment without foliar application."

No such treatment was described in the material and method section.

344-348. lines: "Superoxide dismutase (SOD) activity showed a declining trend with decreasing exposure time compared to the other enzymes. The highest SOD activity was related to the foliar application of CS-SA NCs (6 hours before cold treatment) under 4-hour cold stress. The lowest SOD activity was traced in 16-hour cold treatment without foliar application (Fig. 3)."
Decreasing trend with decreasing exposure time, while the author writes the following sentence in reverse.

lines 368-370: "A significant positive correlation was observed among EL, GSSG, MDA, H2O2, and DHA. Electrolyte leakage negatively correlated with enzymatic antioxidants, AsA, AsA/DHA, GSH, and GSH/GSSG."
At what level is the result significant?

427-429. lines: "Sairam and Saxena (Sairam & Saxena 2000) observed that the membrane stability index was reduced in wheat plants’ tolerant and susceptible cultivars under drought stress. Salinity stress
increased membrane permeability in other plants (Aazami et al., 2021b)."
These references are irrelevant to the topic. Please remove them.

lines 451-453: "Drought stress is involved in protein metabolism in grapes. Amino acid levels often increase under drought stress. This may be due to the re-synthesis or the breakdown of proteins."
These references are irrelevant to the topic. Please remove them.

539-540. lines: "Still, with increasing NaCl concentration, the amount of GSH significantly decreased (Yildirim et al., 2004).
Please remove this reference as well.

I support the publication of the manuscript after the suggested changes and corrections are made.

---

## Round 0.2 · accepted · Accept

· Academic Editor

Accept

Dear Dr. Mlcek, I am pleased to inform you that this article has been accepted for publication.

For instance:

L 40 “various” (spelling error) L 97 “NCs” -- this abbreviation needs to be written out upon first use here.

L 269 “performed using MSTAT-C ver 2.1” - Please cite a reference for this software and also check the version number. The current version is 8.1 while the earliest version documented on the software webpage is 2.6 released in 1998: https://oncology.wisc.edu/mstat/version/index.html and it appears the these earlier versions are not even functional on computer operating systems available today.

L 552 “leadinf” (spelling error) Figure 5 – needs to be replaced as there is a pop-up window on the screenshot (on graph a). All of the figures seem to be screen shots (poor quality) rather than high quality graphics files needed for publication. Figure 7 is completely illegible and needs to be replaced with a high resolution figure.

Reviewer 2 ·

Basic reporting

The authors offer an alternative in the manuscript to help grapes adapt to cold climate conditions. For this reason, raising the topic is particularly timely.

Experimental design

The planned experiment basically meets the requirements of the field, and the authors have corrected methodological errors.
The quality of the evaluation of the results has improved; it is logical and understandable.
The figures and tables are clear, well-structured, interpretable, and suitable for presenting the desired results.

Validity of the findings

The discussion subsection is clearly and well-formulated, supports the results, and the authors compare their own results with relevant literature references. Correcting the objections, the authors have included more relevant literature references in the manuscript, thus better supporting their results.

Additional comments

The authors have made the requested corrections and changes. This has significantly improved the quality of the manuscript.